# Molecular and Metabolic Subtypes Correspondence for Pancreatic Ductal Adenocarcinoma Classification

**DOI:** 10.3390/jcm9124128

**Published:** 2020-12-21

**Authors:** Pilar Espiau-Romera, Sarah Courtois, Beatriz Parejo-Alonso, Patricia Sancho

**Affiliations:** Translational Research Unit, Hospital Universitario Miguel Servet, IIS Aragon, 50009 Zaragoza, Spain; pespiau@iisaragon.es (P.E.-R.); scourtois@iisaragon.es (S.C.); bparejo@iisaragon.es (B.P.-A.)

**Keywords:** PDAC, pancreatic cancer, glycolysis, lipid metabolism, classification, stratification

## Abstract

Pancreatic ductal adenocarcinoma (PDAC), the most common form of pancreatic cancer, is an extremely lethal disease due to late diagnosis, aggressiveness and lack of effective therapies. Considering its intrinsic heterogeneity, patient stratification models based on transcriptomic and genomic signatures, with partially overlapping subgroups, have been established. Besides molecular alterations, PDAC tumours show a strong desmoplastic response, resulting in profound metabolic reprogramming involving increased glucose and amino acid consumption, as well as lipid scavenging and biosynthesis. Interestingly, recent works have also revealed the existence of metabolic subtypes with differential prognosis within PDAC, which correlated to defined molecular subclasses in patients: lipogenic subtype correlated with a classical/progenitor signature, while glycolytic tumours associated with the highly aggressive basal/squamous profile. Bioinformatic analyses have demonstrated that the representative genes of each metabolic subtype are up-regulated in PDAC samples and predict patient survival. This suggests a relationship between the genetic signature, metabolic profile, and aggressiveness of the tumour. Considering all this, defining metabolic subtypes represents a clear opportunity for patient stratification considering tumour functional behaviour independently of their mutational background.

## 1. Introduction

Pancreatic ductal adenocarcinoma (PDAC) is considered one of the most aggressive solid malignancies. It represents the third cause of cancer-related deaths in industrialised countries today [1] and it is predicted to become the second by 2030 [2]. Despite progress in the understanding of the molecular and genetic basis of this disease, five-year survival rates have remained below 10% after diagnosis and one-year survival occurs in only 28% of cases [3].

The reasons for the poor prognosis of this disease include bad accessibility to the organ, absence of distinct symptoms and high rate of metastasis, occurring in about 50% of patients [4]. On the other hand, there are no reliable biomarkers approved for early diagnosis [5].

Currently, the only available curative option for PDAC patients is surgical resection followed by adjuvant chemotherapy [4]. However, this occurs in a minority of patients, since 80–90% of them are diagnosed with advanced disease when the tumour is not resectable [6,7]. In addition, the heterogeneity and plasticity of PDAC tumours lead to chemoresistance [8]. In fact, several phase III trials of chemotherapy agents or targeted therapies effective in other malignancies have failed to benefit unselected PDAC populations. In addition, with the exception of the rare subset of mismatch repair-deficient tumours, checkpoint inhibitors have failed to show efficacy in metastatic patients.

Under such circumstances, the solution possibly lies in early detection and proper classification of patients [4]. Patient stratification has become an invaluable tool for the clinical management of cancer patients, providing diagnostic and prognostic information and, crucially, guiding therapeutic decisions, especially when targeted therapies for a specific mutation or subgroup are available. In fact, stratification had a key role improving survival rates in diseases such as breast cancer. However, patient stratification based on histomorphological or molecular features in PDAC has proved challenging, severely delaying the identification of such targeted therapies. This, together with the undruggable nature of mutated K-RAS (although clinical trials are underway to inhibit the mutation K-RAS^G12C^), limits PDAC treatment to ineffective conventional chemotherapy. Genomic studies have revealed subtypes of PDAC based on their molecular features, but diverse studies have reported different classification systems incompletely overlapping. Interestingly, increasing efforts are being made in different directions. In fact, recent data suggest that PDAC tumours could also be classified attending to their metabolic phenotype.

In this review, we describe and analyse the correspondence of most of the published molecular and metabolic signatures of PDAC and propose a modified metabolic signature that could stratify patients according to metabolic needs independently of the mutational load.

## 2. PDAC Mutational Profile

Many studies have been conducted on PDAC samples to determine molecular aberrations at DNA, RNA, protein and epigenetic levels [9]. In general terms, mutations in individual genes comprise 10 main pathways: DNA damage repair (*TP53* or *BRCA2*), cell cycle regulation (Cyclin-Dependent Kinase Inhibitor 2A, *CDKN2A*), Transforming growth factor-beta (TGF-β) pathway (*SMAD4*), NOTCH and WNT signalling, chromatin regulation, Extracellular Signalling Regulated Kinases (ERK)-Mitogen-Activated Protein Kinases (MAPK) signalling (*K-RAS*, *B-RAF*), axonal guidance and RNA processing [10,11].

Altogether, *K-RAS* gain of function, accounting for up to 90% of PDAC cases, and inactivation of the tumour suppressor genes *p16/CDKN2A*, *SMAD4* and *TP53* contribute to PDAC formation, maintenance, progression and, ultimately, metastasis onset. Occasionally, the mutations occur in other components of the pathway in which they are involved [9,12,13]. In addition, genome-wide and exome-wide analyses have identified a long list of less frequent alterations in genes related to axon orientation or DNA damage repair, such as *BRCA1* or *BRCA2* [12,14,15]. Moreover, most PDACs show complex patterns of chromosomal rearrangement [16,17].

Some findings have suggested a linear pathway of evolution from normal cells to PDAC cells, in which not all mutations occur at the same time. Telomere shortening is considered the trigger of pancreatic tumorigenesis by inducing genetic instability, coupled with a mutation in the *K-RAS* gene. Surviving cells are then altered by loss of *CDKN2A* function. In third place, the tumour suppressor genes *TP53* and *SMAD4* are inactivated, leading to intermediate stages called pancreatic intraductal neoplasias (PanINs) and, ultimately, full-blown PDAC [10,18]. However, it seems that in some cases, *CDKN2A*, *TP53* and *SMAD4* mutations can occur in a single event, which is called “punctuated equilibrium” [11].

### 2.1. K-RAS

*K-RAS* is one of the three mutationally activated forms of the Ras protein, together with *H-RAS* and *N-RAS*. It is the most represented one among all cancers bearing RAS mutations, and the one exclusively mutated in PDAC. Moreover, it is known to be the main driver mutation for this disease, since *K-RAS* is mutated in >90% of the cases, thus making PDAC the most RAS-addicted of all cancer types [19]. However, *K-RAS* activation alone is not sufficient for PDAC development [20,21].

Ras proteins are small GTPases that act as signalling switches regulating cell proliferation, survival, differentiation and inflammation, by activating the MAPK module. In a normal scenario, Ras is found in its inactive form coupled to guanosine diphosphate (GDP) and, upon activation by diverse extracellular stimuli, Ras becomes active by coupling guanosine triphosphate (GTP), thus switching on downstream effectors such as ERKs, Jun Amino-Terminal Kinases (JNKs), Stress-Activated Protein Kinases (p38/SAPKs) and PI3K/PDK1/AKT, amongst others [22,23]. In PDAC and other cancers bearing oncogenic *K-RAS* mutations, this protein is constitutively active in a GTP-bound state. PDAC-associated *K-RAS* alterations mainly harbour missense mutations that depend on allele dosage for tumorigenesis and metastasis onset, and contribute to PDAC evolution from PanINs to final cancer formation with the loss of other key PDAC-driving players [24].

### 2.2. p16/CDKN2A

Along the evolutionary course of PDAC onset, *K-RAS* mutation is followed by the occurrence of genetic alterations in the tumour suppressor gene *CDKN2A* in 95% of PDAC patients [24]. Tumour suppressor genes are important to restrain aberrant cell proliferation in the context of oncogenic signalling and, to that end, these genes rely on different strategies such as cell cycle arrest, apoptosis or senescence induction. Specifically, the gene *CDKN2A*, encoding for P16INK4a and P14RF proteins, shows a dual mechanism: P16INK4a inhibits Cyclin Dependent Kinases (CDK) 2 and 4, thus preventing cell cycle to progress into S-phase, while P14RF triggers P53 by inhibiting its inhibitor, Mouse double minute 2 homolog (MDM2), thereby inducing growth arrest and apoptosis [25].

### 2.3. TP53

P53, the protein encoded by the *TP53* gene, is known to be the “genome guardian” due to its essential role upon chromatin damage. Amongst its main functions, we find cell cycle blockade and activation of DNA repair enzymes. Ultimately, P53 may lead the cell to a senescence state and/or apoptosis when DNA damage is no longer reparable. Genetic inactivation of *TP53* is present in up to 75% of all PDAC cases and appear in advanced PanINs after loss of *CDKN2A*. Loss of functional P53 leads to an uncontrolled state of proliferation [25]. Importantly, mutant P53 may contribute to metastatic progression of K-RAS-driving PDAC-bearing mice, highlighting a novel role of this transcription factor in late stages of PDAC beyond the malignant transformation of PanINs into PDAC [26,27].

### 2.4. SMAD4

*SMAD4* serves as the central mediator of TGF-β pathway and it is known to be the fourth and, together with *TP53*, ultimate driver for PDAC initiation [28]. However, the role of SMAD4 in the pathogenesis of PDAC is complex and its final results are context-dependent. On the one hand, the TGF-β pathway is a well-accepted epithelial-to-mesenchymal transition (EMT) inducer [29], and also in PDAC [25], thus highlighting its pro-tumorigenic potential. Conversely, Smad4 signalling in the KPC (LSL-KrasG12D/+;LSL-Trp53R172H/+;Pdx-1-Cre) mouse PDAC model mediates a tumour suppressive process known as “lethal EMT”. In this scenario, Smad4 triggers apoptosis through repression of the gastrointestinal lineage-master regulator Klf5 and, consequently, inhibits PDAC progression induced by Klf5/Sox4. Consequently, the loss of SMAD4 tumour suppressive function may contribute to pancreatic oncogenesis through the TFG-β canonical pathway [30].

## 3. PDAC Molecular Signatures

Besides purely mutational analyses, a considerable number of genomic and transcriptomic studies in patient samples have identified the existence of PDAC subtypes with prognostic and biological relevance, as detailed below.

Collisson et al. [31] reported the first exhaustive transcriptomic analysis on clinical samples, and based their classification on gene expression and molecular profiling. PDAC tissues were classified into “classical”, “quasi-mesenchymal” and “exocrine-like” (Table 1), with different prognoses and responses to selected therapies. The classical subtype showed high expression of epithelial and adhesion-associated genes, such as the transcription factor *GATA6*, and favourable prognosis regarding survival after PDAC resection. On the other hand, the quasi-mesenchymal subtype exhibited high expression of mesenchymal-associated genes, was relatively less dependent on K-RAS than the classical one, and, importantly, was associated with poor prognosis. Finally, the exocrine-like subtype revealed high expression of digestive exocrine enzyme genes. However, the last subtype was not found in cancer cell lines, raising concerns about its specificity [32].

Moffitt et al. [33] performed transcriptional analysis to classify PDAC tumours into two subtypes within two different tissues (Table 1): tumour and stroma. Based on tumour cells, PDAC samples were classified into “classical” and “basal” subtypes. The classical group was enriched in *SMAD4* and *GATA6*, while the basal subtype was molecularly similar to basal tumours in bladder and breast cancer. Although classical tumours showed better prognosis, patients with basal tumours responded better to adjuvant therapy. On the other hand, the stroma was divided into “normal” and “activated” subtypes. The normal stromal group showed high levels of markers for pancreatic stellate cells, smooth muscle actin, vimentin and desmin. However, the activated stromal subtype was defined by a complex set of genes associated with macrophages, tumour promotion and fibroblast activation; this described an activated pro-tumorigenic inflammatory stromal response with poor prognosis. Comparing their signature with Collisson’s [31], Moffitt et al. found an overlap between both classical subtypes, while genes in the quasi-mesenchymal subtype seemed a mixed selection of genes from basal and stromal subtypes (Table 2) [33].

Later, Bailey et al. [10] defined new molecular subtypes of PDAC based on comprehensive integrated genomic analysis and RNA expression profiles: “squamous”, “pancreatic progenitor”, “aberrantly differentiated endocrine exocrine” (ADEX) and “immunogenic” (Table 1). The squamous subtype was linked to hypermethylation and down-regulation of genes determining endodermal identity in pancreas, with poor prognosis in patients. It was also enriched in *TP53* and *KDM6A* mutations and up-regulation of the transcriptional network *TP63*Δ*N*. PDAC progenitor-type tumours expressed genes involved in early development of the pancreas, such as *PDX1*. The ADEX class overexpressed transcriptional pathways present in K-RAS activation and in late stages of pancreatic development and differentiation. Lastly, the immunogenic subtype had pancreatic progenitor features, but it was related to acquired immune suppression pathways (*CTLA4* and *PD-1*) and strong immune infiltration (B and T cells). When they compared their classification with earlier studies, three of their classes directly overlapped with Collisson’s groups [31]: Collisson’s classical, quasi-mesenchymal and exocrine-like subtypes matched Bailey’s pancreatic progenitor, squamous and ADEX groups, respectively (Table 2). In addition, 50% of the squamous subtype tumours were included in Moffitt’s basal subgroup [33].

In 2018, Zhao et al. [34] conducted a retrospective meta-analysis of complete transcriptome data from patients with PDAC (Table 1). They obtained six different subtypes grouped in tumour- (L1, L2 and L6) and stromal-specific (L3, L4 and L5) subtypes. L1, L2 and L6 were all enriched in metabolic genes, as detailed in Section 5. Besides metabolism, the subtypes were differentially enriched in other routes: L2 was related to cell proliferation and epithelium genes, such as *CDKN2A*; L3 had increased regulation of collagen and extracellular matrix (ECM)-associated genes; L4 had an immune profile; L5 was involved in neuroendocrine and insulin-related pathways; L6 showed activity of digestive enzymes. Each expression profile was related to clinical data: epithelium and proliferation in L2 showed the poorest prognosis, immune and neuroendocrine profiles at L4 and L5 showed the best outcomes. Considering the above stratifications of PDAC, Zhao et al. found that L1 and L6 were very similar to the classical Moffit’s subtype, while L2 approached their basal subtype (Table 2). L1, L2 and L6 were close to the activated subtype. In Bailey’s dataset, L1 and L4 approximated the immunogenic subtype, L2 the squamous, L3 the pancreatic progenitor and L6 resembled the ADEX subtype. Finally, L1 and L3 were similar to Collison’s classical subtype, L2 approached the quasi-mesenchymal and L6 related to the exocrine-like one.

Lomberk et al. [35] identified epigenomic landscapes of PDAC subtypes by performing an integrative analysis of genome-wide Chromatin Immunoprecipitation-sequencing (ChIP-seq) on multiple histone modifications, as well as RNA-sequencing (RNA-seq) and DNA methylation studies (Table 1). They reported two subtypes: “classical” and “basal”, which correlated with clinical parameters. Super enhancer mapping coupled with Transcription Factor (TF) binding motif and up-regulation analysis showed that classical tumours were related to TFs involved in development of the pancreas, metabolic regulators and K-RAS signalling, while basal tumours correlated with proliferative and EMT TF nodes. The basal samples were linked to a more aggressive phenotype than the classical ones.

Maurer et al. [36] used Laser Capture Microdissection (LCM) to separate stromal and epithelial compartments of PDAC bulk tumour samples, and confirmed using machine learning techniques that stromal contamination may interfere with PDAC classification (Table 1). In addition, they described two epithelial subtypes of PDAC that correlated with the classical and basal subtypes from Moffit et al., with the basal subtype having a poorer prognosis than the classical (Table 2). Moreover, they identified two stromal subtypes, an “immune-rich” group characterised by high immune and interleukin levels, and an “ECM-rich” group, associated with extracellular matrix pathways. The ECM-rich subtype appeared to have decreased survival compared with the immune-rich one. Finally, they found an association of the epithelial and stromal subtypes in which the basal epithelium subtype and the ECM-rich stroma were linked.

In 2020, Dijk et al. [37] conducted an unsupervised consensus clustering and identified four molecular subtypes: “secretory”, “epithelial”, “compound pancreatic” and “mesenchymal” (Table 1). The secretory subtype showed enrichment in endocrine and exocrine pathways of the pancreas. Tumours of the epithelial subtype were characterised by up-regulation of the *MYC* oncogene and high expression of mitochondrial components and ribosome signature. The mesenchymal subtype had high levels of *K-RAS* transcription and was enriched in pathways related to EMT, stromal signalling and TGF-β. The compound pancreatic subtype presented similar characteristics to the mesenchymal subtype but, in addition, was enriched in endocrine functions. The secretory and mesenchymal subtypes presented worse prognosis than the epithelial and compound pancreatic ones. Dijk et al. found an interconnection between these subtypes and those described by Collison, Bailey and Moffit (Table 2). For example, the secretory subtype correlated significantly with Collison’s exocrine-like, Bailey’s ADEX, and Moffit’s basal subtypes. The epithelial subtype shared characteristics with the Collison’s and Moffit’s classical subtypes, but also with the Bailey´s pancreatic progenitor and squamous subtypes. The compound pancreatic group was similar to Collison’s exocrine, Bailey’s ADEX and Moffit’s classical. Lastly, the mesenchymal subtype correlated with Collison’s quasi-mesenchymal, Bailey’s squamous and Moffit’s basal.

Chan-Seng-Yue et al. [38] performed whole genome sequencing and both bulk and single-cell RNAseq analyses on laser capture microdissected tumours from more than 200 late-stage patients. They identified three major subtypes: “classical”, “basal-like” and “hybrid” (Table 1). Although the classical and basal-like subtypes were fairly overlapped with the previous classifications by Moffit, Bailey and Collison (Table 2), they found that these subtypes could be subdivided into A and B subclasses with differences in their response to chemotherapy, aggressiveness and disease stage. For example, basal-like A tumours are highly chemoresistant and feature a high squamous signature enriched in metastatic tumours, while basal-like B tumours correspond to a low squamous signature present in resectable tumours. Importantly, they found that classical and basal-like programs co-exist within a tumour and demonstrated that molecular subtypes are linked to a specific copy number aberrations in genes such as mutant *K-RAS* (basal-like subtype) and *GATA6* (classical subtype).

Finally, Nicolle et al. [39] proposed a molecular gradient classification to stablish a PDAC transcriptomic signature that could better represent the existence of intermediate cellular phenotypes between classical and basal-like subtypes (Table 1). In this study, they graded Patient-Derived Xenografts (PDXs) according to different molecular levels of differentiation and confirmed that the higher expression of genes linked to the classical PDAC subtype (*GATA6*) was correlated with increased differentiation of PDX samples, while lower expression of genes was linked to the basal-like subtype. They also found that the higher the differentiation of the tumour, the better the prognosis and response to chemotherapy.

Interestingly, although partial overlap amongst the different signatures and molecular subtypes can be found in these studies, recent works using single-cell transcriptomics demonstrated that several subtypes can co-exist within a tumour [38,40], further underlining the high level of heterogeneity present in PDAC tumours. Undoubtedly, more efforts in this direction are required to further dissect the complexity of these tumours, as well as to decipher the interplay of the different subpopulations during disease progression.

## 4. Metabolic Reprogramming in PDAC

At the histological level, one of the most notable characteristics of PDAC is its dense stroma, being up to 90% of the tumour volume. Its main features are extensive fibrosis, lack of vascularisation, hypoxia and immune infiltration. Hypoxia is associated with increased cancer cell proliferation, survival, EMT, invasiveness and metastasis [41].

The lack of vascularisation not only causes hypoxia in the tumour, but also causes metabolic stress due to nutrient deprivation. As a result, tumour cells undergo the so-called “metabolic reprogramming”, an updated hallmark of cancer [32,42]. Cancer cells increase nutrient acquisition along with enhanced flow through anabolic pathways. This leads to increased glycolysis and glucose transport, high glutamine consumption, lipid and amino acid biosynthesis and maintenance of redox homeostasis. In addition, recycling of cellular components also occurs through autophagy, which degrades macromolecular complexes and organelles into individual metabolites [41,43,44,45].

### 4.1. Warburg Phenotype

The major example of metabolic reprogramming is higher glucose consumption. The role of glucose metabolism in cancer was well defined by Otto Warburg back in the late 1920s, leading to be considered one of the hallmarks of cancer [46]. Glycolytic flow is precisely controlled to fulfil rapid proliferative and synthetic needs. Unlike normal cells, tumour cells have high levels of glycolysis, even in the presence of oxygen and reduced mitochondrial function, leading to a state called “aerobic glycolysis”, also called the “Warburg effect”. On the other hand, the “reverse Warburg effect” describes a two-compartment model in which cancer cells induce aerobic glycolysis in the stromal cells, whose glycolysis end-products are transferred to the cancer cells to feed mitochondrial oxidative phosphorylation (OXPHOS). This allows tumours to respond to variations in nutrient availability and to optimise cell proliferation and growth [47]. Interestingly, this two-compartment model can also be applied considering the functional heterogeneity of cancer cells in PDAC: glycolytic differentiated tumour cells could provide substrates to oxidative cancer stem cells (CSCs) [48].

A hypothesis for metabolic rewiring towards enhanced glycolysis over the reduction of mitochondrial oxidation as a source of ATP in PDAC is the presence of a dense desmoplastic stroma, which basically impedes neovascularisation. This creates a hypoxic microenvironment in which oxygen and nutrients are limited [49]. This state creates a positive feedback loop by which, on the one hand, pancreatic cancer cells feel a selective pressure under the hypoxic and nutrient shortage where only the most aggressive populations will remain. On the other hand, under such stress-driven situations, pancreatic cancer cells are forced to modify their metabolism in order to cope with their bioenergetic demands for PDAC progression, expansion and dissemination through the blood vessels towards less scarce environments [50]. Indeed, a recent study demonstrated that glycolysis and hypoxia signatures correlate in PDAC and that Prolyl 4-Hydroxylase subunit Alpha 1 (P4HA1), a critical enzyme involved in collagen synthesis, controls glycolysis through HIF1α stabilisation [51]. Moreover, gene expression arrays of metastatic PDAC revealed a glycolysis-based signature characterised by increased expression of many glycolytic enzymes [52], highlighting the importance of glycolytic metabolism in PDAC progression. In fact, Liu et al. recently demonstrated that the EMT-related gene *SNAIL* was able to induce a migratory phenotype in PDAC cell lines by promoting mesenchymal-related genes expression along with enhanced glucose uptake and lactate production [53].

Overall, the glycolytic state is characterised by an increased expression of glycolytic enzymes and glucose and lactate transporters, such as Glucose Transporter 1 (GLUT1), and Monocarboxylate Transporters 1 and 4 (MCT1, MCT4) [54,55]. Specifically, the overexpression of these membrane transporters leads to an enhanced glucose scavenging from the hypovascularised tumour microenvironment, which results in increased glucose availability in the cancer cell as well as a better balance of the glucose pathway in order to keep glycolysis at high rate. On the one hand, GLUT1 is an ATP-independent glucose transporter that enables glucose transference from a high-gradient extracellular compartment to low-gradient cytoplasmic compartment. Its expression dosage has been reported to be associated with PDAC progression from low- to high-grade pancreatic preneoplastic lesions when compared to normal pancreas [54]. On the other hand, MCT1 and MCT4 are proton-coupled symport transporters with higher affinity for lactate efflux. These transporters implicated in glucose homeostasis are required for cancer cells to neutralise intracellular acidification due to the increased glycolytic rate and have been reported to be overexpressed in PDAC. Kong et al. showed that inhibition or knockdown of MCT resulted in an inhibited lactate flux. Interestingly, they also demonstrated that these transporters are implicated in PDAC cell lines invasiveness, thus highlighting once again the implications of glucose homeostasis in pancreatic cancer [56].

Importantly, the up-regulation of most of these genes is mediated by PDAC driver mutations on *K-RAS* and *TP53*. As reviewed by Bryant et al. [57], oncogenic K-RAS enhances the expression of many glycolytic enzymes such as GLUT1, Hexokinase 1 and 2 (HK1, HK2) and Lactate Dehydrogenase A (LDHA), thus increasing glycolytic flux. This metabolic reprogramming towards glycolysis contributes to an enhanced survival of glycolytic PDAC cell lines in the presence of low levels of glucose. Moreover, another study based on transcriptome and metabolome analyses showed that mutant K-RAS in advanced PDAC mouse models is necessary for an enhanced glucose uptake [57]. This study also revealed that aberrant K-RAS is implicated in glucose metabolism intermediates channelling into different anabolic pathways, such as the hexosamine biosynthesis pathway (HBP) and pentose phosphate pathway (PPP), thus proving that glucose metabolism is necessary to fuel anabolic branches of PDAC metabolism to provide the cancer cells with building blocks for its increased proliferation demands [58]. Moreover, the enhanced expression of the glycolytic enzymes was demonstrated to be related to bad PDAC prognosis, invasiveness and metastases onset [59].

### 4.2. Lipid Metabolism in PDAC

PDAC tumours are also highly dependent on lipid metabolism [60,61] and, in fact, a high fat diet was shown to support tumour growth in murine models [60]. On the one hand, fatty acids (FA) can be provided exogenously by the absorption of extracellular lipids (from diet, liver synthesis or adipose tissue). For example, cancer-associated adipocytes can provide adipokines and lipids to cancer cells [62], increasing pancreatic cancer cell aggressiveness [63]. The exogenous FA uptake requires the presence of the transporter CD36 and FA-binding proteins (FABPs). CD36 can also influence gemcitabine resistance in PDAC, by regulating anti-apoptosis proteins [64,65]. Unsurprisingly, PDAC patients with high CD36 expression have lower overall survival and recurrence-free survival rates than patients with low expression. In this context, CD36 could be considered as an unfavourable prognosis factor and the use of anti-CD36 strategies in association with conventional chemotherapies could represent a promising therapeutic approach [64,66].

On the other hand, pancreatic cancer cells can synthesise de novo lipids through the lipogenesis process, producing more than 90% of the triacylglycerol-FA. This process uses the mitochondrial citrate produced from the tricarboxylic acid (TCA) cycle fuelled by glucose and/or glutamine as carbon sources [67]. Lipogenic enzymes are often overexpressed in PDAC cells; for example Fatty Acid Synthase (FASN) is particularly prominent and associated with poor prognosis [61,68,69,70]. The pharmacological inhibition of this enzyme reduces stemness features and gemcitabine resistance in pancreatic cancer cells [71]. The produced triacylglycerol molecules are stored in lipid droplets and a correlation was established between the accumulation of lipid droplets and tumour progression and aggressiveness [72]. Likewise, an elevated lipid synthesis correlates with CSC properties and survival in different types of cancer [72]. Thus, cancer cells acquire FA through either lipid uptake or *de novo* lipogenesis, and activate intracellular lipolysis to mobilise the FA stocks.

FA sustain three requirements of PDAC development and cancer cells in general: cell membrane formation, biosynthesis of signalling molecules and lipid-derived messengers, and energy production. First of all, lipid synthesis is an important requirement of highly proliferative cancer cells to sustain membrane formation [72]. From *de novo* synthesis, saturated or monounsaturated FA modulate membrane fluidity and form more dense membrane layers that may reduce the uptake of drugs and contribute to therapy-resistance [73]. Moreover, lipids are implicated in signal transduction in two different ways: by building lipid rafts modulating protein recruitments and interactions, as well as by formation of lipidic signalling molecules. This is the case for phosphatidylinositol-3,4,5-trisphosphate [PI(3,4,5)P3], able to activate the protein kinase B/AKT and stimulate cell proliferation and survival [73]. Finally, FA represent an important source of energy in non-glycolytic tumours, using the mitochondrial β-oxidation (fatty acid oxidation, FAO) to produce ATP [72,73]. In that context, Luo et al. demonstrated that the use of etomoxir, an inhibitor of FAO by blocking the entrance of FA in the mitochondria via Carnitine Palmitoyl Transferase 1A (CPT1A), can restore the sensitivity of pancreatic CSCs to gemcitabine by inducing an energy crisis in those cells [74]. These data suggest that CPT1A is an important actor of the cancer metabolism reprogramming and could represent an attractive therapeutic target, and highlight how this process is important for the cancer cell to supply ATP under energy stress.

Additionally, PDAC cells are also highly dependent on cholesterol, as it contributes, for example, to the formation of lipid rafts, thus modulating the recruitment of key oncogenes receptor such as the Epidermal Growth Factor (EGF)-receptor and regulating survival pathways. Cancer cells can increase their content through either synthesis (through mevalonate pathway) or endocytosis mediated by low-density lipoproteins (LDL) and LDL Receptor (LDLR). In PDAC patients, cholesterol biosynthesis is associated with a more differentiated phenotype (classical subtype), while high LDLR expression correlates with a higher risk of tumour recurrence. Interestingly, inhibition of cholesterol synthesis induced a mesenchymal phenotype [75] while blocking cholesterol uptake via the knock-down of LDLR was able to sensitise PDAC cells to chemotherapy [76,77].

Taken together, all these reports prove the potential of lipid metabolism targeting in order to sensitise PDAC cells to chemotherapy. However, some antitumourigenic effects of specific FA like palmitic and stearic acids were reported, and they would be able to trigger apoptosis and inhibit proliferation of pancreatic cancer cells [78]. Additionally, more research is required to fully understand the crosstalk of the different metabolic pathways. As an example, branched-chain amino acids (BCAAs) were able to sustain pancreatic cancer growth by regulating lipid metabolism [79].

### 4.3. Amino Acid Metabolism in PDAC

In their nutrient-deprived environment, PDAC cells also face the lack of amino acids (AAs) and use different processes to counteract this phenomenon and support their metabolic needs. First of all, several studies demonstrated an association between elevated plasma BCAAs levels and pancreatic cancer risk [80,81]. As increased consumption of BCAAs may occur about 10 years before PDAC diagnosis, plasma AAs concentrations can be considered as pre-diagnostic and diagnostic markers [82]. Moreover, the significant differences observed between the different PDAC stages make AAs good candidates to improve early diagnosis and patient stratification [82]. For example, analysis of AAs plasma levels in extended cohorts led to identifying three natural BCAAs with significant elevated concentrations: leucine, isoleucine and valine, metabolised via common pathways [80,83].

The close microenvironment represents another source of AAs to feed PDAC cells. For example, stromal cells, such as pancreatic stellate cells, secrete alanine, which is assimilated by PDAC cells to support their glutamine and glucose metabolism [84,85]. In another way, cancer-associated fibroblasts (CAFs) present an up-regulated BCAAs catabolism and are able to fuel PDAC cells with branched-chain α-ketoacid (BCKA), thanks to their elevated BCAT1 (Branched chain amino acid transaminase 1) activity [86]. It was also demonstrated that PDAC cells can directly catabolise extracellular collagen to produce proline and fuel the TCA cycle under restrictive nutrient conditions [87]. These exchanges between PDAC cells and extracellular milieu (plasma or closed microenvironment) are dependent on the expression of transporters such as the L-type Amino Acid Transporter 1 (LAT1) or the Cystine/Glutamate Exchanger (SLC7A11/xCT). For that reason, the expression of aa transporters is also associated with low prognosis and chemoresistance [88,89].

Additionally, indirect sources of AAs also contribute to PDAC development. In general, a whole-body protein breakdown occurs in the course of the disease, since PDAC cells are able to capture of external proteins, such as albumin, through the macropinocytosis process, which combines endocytosis and protein degradation (lysosomal or proteolytic degradation). This extracellular protein catabolism represents an important source of AAs, including glutamine, that sustains the central carbon metabolism. Indeed, although glucose is the dominant energy fuel for most cancers in vivo, it has been suggested that ATP generation relies on glutamine carbons in vitro, leading to glutamine addiction. Furthermore, glutamine plays other important roles in PDAC cells: (1) in lipid biosynthesis, (2) as a nitrogen donor for AAs and nucleotide biosynthesis, (3) as a carbon substrate for the anaplerosis of the mitochondrial TCA cycle, (4) in redox balance by participating in glutathione biosynthesis and generating NADPH [60,90]. PDAC cells metabolise glutamine through a non-canonical pathway driven by *K-RAS* or *MYC* oncogenes in which transaminases, such as Aspartate Aminotransferase, are essential [91].

## 5. PDAC Metabolic Signatures

Although initially it was thought that all PDAC tumours carried similar changes in metabolism, the existence of subtypes with specific metabolic requirements has become apparent lately. This section summarises and analyses reports that either directly identify metabolic subtypes in PDAC or their existence can be inferred from the signatures classifying each subtype.

Daemen et al. [92] identified three different metabolic subtypes in PDAC cell lines through metabolite profiling, further confirmed by transcriptional analysis: “slow-proliferating”, “glycolytic” and “lipogenic” (Table 3). The slow-proliferating subtype was defined by reduced cellular proliferation and low levels of AAs and carbohydrates. The glycolytic subtype showed high levels of gene expression and metabolites from the glycolytic, pentose phosphate and serine pathways. In contrast, the lipogenic group was characterised by sets of lipogenic genes and metabolites involved in the synthesis of cholesterol and lipids and mitochondrial OXPHOS. Interestingly, while glutamine contributed to TCA metabolites in glycolytic cell lines, the lipogenic ones used glucose to replenish the TCA cycle, which was accompanied by increased oxygen consumption and mitochondrial content. Finally, they created a signature defined by the expression ratio of the glycolytic gene *ENO2* (neuron-specific enolase) and several lipid genes, further validated in 200 non-pancreatic cancer cell lines. Interestingly, the authors observed a correlation among their metabolic subtypes and Collison’s molecular subtypes [31]: Daemen’s lipogenic subtype is associated with the classical subtype, while the glycolytic one is related to the quasi-mesenchymal subtype (Table 4). Accordingly, and in contrast to the lipogenic subtype, the glycolytic subtype is very aggressive, grows rapidly, produces metastases and generally does not respond to regular chemotherapy [92].

As mentioned before, the L1, L2 and L6 PDAC subtypes described by Zhao et al. [34] were related to metabolism (Table 3), while the L1 tumours up-regulated glycolytic and lipogenic genes and the L2 subtype enriched glycolysis gene sets and down-regulated lipid metabolism ones. Finally, L6 up-regulated genes associated with protein metabolism and digestive enzyme activity. We can find similarities between Zhao’s and Daemen’s classifications: L2 would correspond to Daemen’s glycolytic subtype, associated with poor prognosis in clinical samples, while L1 would constitute a mixed group with characteristics of both glycolytic and lipogenic groups (Table 4).

The work by Lomberk et al. [35] analysing the epigenetic landscape of PDAC samples identified super-enhancers and super-enhancer-associated TFs regulating metabolic nodes (Table 3). Indeed, the classical subtype is associated with the up-regulation of TFs modulating lipid metabolism (PPAR), while MYC, a known regulator of PDAC glycolytic phenotype [47], likely controls the basal subtype.

Maurer et al. [36] conducted analyses of gene set variance in different published cohorts (TGCA, ICGC, UNC and CUMC), applying transcriptional deconvolution to identify the genes specifically associated with the epithelial compartment (Table 3). Interestingly, they confirmed the correlation between the classical subtype and lipid metabolism in the diverse datasets, highlighting cholesterol biosynthesis and retinol metabolism (Table 4).

A recent study by Karasinska et al. [93] analysed the expression of genes related to glycolysis and cholesterol synthesis in clinical samples (Table 3) to establish four metabolic subgroups of PDAC: “quiescent”, “glycolytic”, “cholesterogenic” and “mixed”. While the quiescent group had poor metabolic activity, the glycolytic and cholesterogenic subtypes enhanced one of these pathways. The glycolytic subtype was characterised by the amplification of *K-RAS* and *MYC* oncogene and showed the lowest expression of the mitochondrial pyruvate carriers *MPC1* and *MPC2*. In addition, the glycolytic group showed poor prognosis, whereas the cholesterogenic one reported the longest median survival. Lastly, the mixed subtype presented high metabolic activity and enrichment in both the glycolytic and cholesterol biosynthesis pathways. These metabolic subtypes were also found in nine additional types of cancer. Karasinska et al. compared their metabolic classification with previous molecular signatures (Table 4): the quiescent group predominantly belonged to Moffit’s classical subgroup, and it showed the highest frequency of ADEX and exocrine-like cases, suggesting that the quiescent group might be involved in the secretion of digestive enzymes; the glycolytic subtype was associated with the basal, squamous and quasi-mesenchymal subgroups, all related with the worst outcome; finally, the cholesterogenic group had the lowest proportion of poor prognosis signatures, but the highest in Bailey’s pancreatic progenitor subtype.

Coming back to the classification of Dijk et al. [37], the epithelial subtype showed an enrichment in mitochondrial and lipid metabolism pathways such as OXPHOS, terpenoid backbone biosynthesis, TCA cycle, steroid biosynthesis and FA degradation (Table 3). This suggests a link between Dijk´s epithelial subtype and Daemen´s and Karasinka´s lipogenic subtypes, all of them correlated to the classical classification (Table 4).

As seen for both molecular and metabolic stratification, all these classifications share subtypes that are consistently present in PDAC. All stratifications comprise a classical subtype with better prognosis corresponding to the lipogenic metabolic subtype. On the other hand, the quasi-mesenchymal, basal or squamous subtypes, which are aggressive, metastatic and undifferentiated tumours, are always related to the glycolytic subtype [32,94]. This strongly suggests an association amongst the genetic signature, metabolic profile and tumour aggressiveness. Such a connection could reflect the transcriptional program of a driver mutation such as the ones found in *K-RAS*, which induces profound changes at the cellular level, including metabolic reprogramming. However, the heterogeneity found in molecular and metabolic subtypes does not necessarily correspond to specific mutations, suggesting an extra layer of complexity provided, for example, by the microenvironment. Thus, defining metabolic subtypes represents a clear opportunity for patient stratification, considering tumour phenotype independently of its mutational background, and opens a new avenue for the identification of targeted therapies for each subgroup.

## 6. Metabolic Phenotypes and Survival in PDAC

As mentioned above, we can define at least two metabolic phenotypes consistently represented across the different analyses, regardless of the main focus of the studies (molecular or metabolic classification): glycolytic and lipogenic subtypes. Interestingly, bioinformatic analyses using TCGA and GTEx databases indicate that genes from both metabolic pathways are significantly up-regulated in tumoural vs. normal tissues (Figure 1). Amongst the genes included in this analysis, we can find representative genes of the metabolic subtypes defined by Daemen et al., Zhao et al. and Karasinska et al. in their respective stratification studies [34,92,93]: (1) for the glycolytic subtype, these are *ENO2* (Daemen, Zhao, Karasinska) and lactate dehydrogenase 1 (*LDH1*) (Karasinska); (2) for the lipogenic subtype, it concerns those involved in the triacylglycerol and cholesterol biosynthesis, such as diacylglycerol O-acyltransferase 1 (*DGAT1*), 7-hydrocholesterol reductase (*DHCR7*), 3-hydroxy-3-methylglutaryl-CoA synthase 1 (*HMGCS1*) and mevalonate (diphospho-) decarboxylase (*MVD*) (Daemen, Karasinska).

Disease-free survival analyses for the representative genes of the metabolic stratification studies mentioned above revealed that, except for *LDH1* and *DHCR7*, none of the individual genes had a significant impact on patient survival (Figure 2a,b). Surprisingly, although not significant, a trend for *ENO2* expression associated with better prognosis was observed (Figure 2a). In fact, previous studies proved that up-regulated *ENO2* contributes to the aggressiveness and poor prognosis of the glycolytic subtype in pancreatic cancer [34,92,93]. Importantly, Daemen et al. proposed a gene expression ratio based on *ENO2* and a lipid signature composed by *DGAT1*, *DHCR7*, *FDFT1*, *HMGCS1* and *MVD* to distinguish between glycolytic and lipogenic metabolic phenotypes for PDAC and other cancer types. Considering the survival analyses in Figure 2a, we analysed the relationship between the enrichment in glycolytic or lipogenic genes and prognosis, using a modified glycolytic/lipogenic gene ratio from Daemen’s study that includes *LDH1* instead of *ENO2*. In this line, disease-free survival decreased significantly when the ratio *LDH1* and any of the lipogenic genes increased; thus, when tumours were glycolytic (Figure 2c). Interestingly, the hazard ratio (HR) combining *LDH1* and *MVD* or *DGAT1* reached values of up to 3.8, facilitating the distinction between the two metabolic subtypes and their impact on prognosis. Although further work in this direction is certainly needed, the use of a simplified signature with the metabolic genes that most contribute to prognosis could streamline patient stratification based on the phenotypic features of the tumours.

## 7. Conclusions

PDAC is a heterogeneous disease that can be classified, according to its metabolic needs, into glycolytic and lipogenic subtypes with different prognosis. Indeed, the glycolytic subtype would be more aggressive and resistant to conventional chemotherapy than the lipogenic one. This metabolic stratification correlates with previous molecular classification systems, as they show subtypes sharing similar characteristics and prognosis. In this context, classification into metabolic subtypes may be better placed for clinical use, since it could provide information on the functional phenotype of the tumour, correlating to aggressiveness, chemoresistance and metastatic abilities independently from its mutational state. The information collected in this review strongly suggests that patient stratification based on metabolic features may bear prognostic value and guide therapeutic decisions in the future, identifying a subgroup of patients with poor prognosis that may be eligible to personalised treatments designed according to metabolic vulnerabilities.

## Figures and Tables

**Figure 1 jcm-09-04128-f001:**
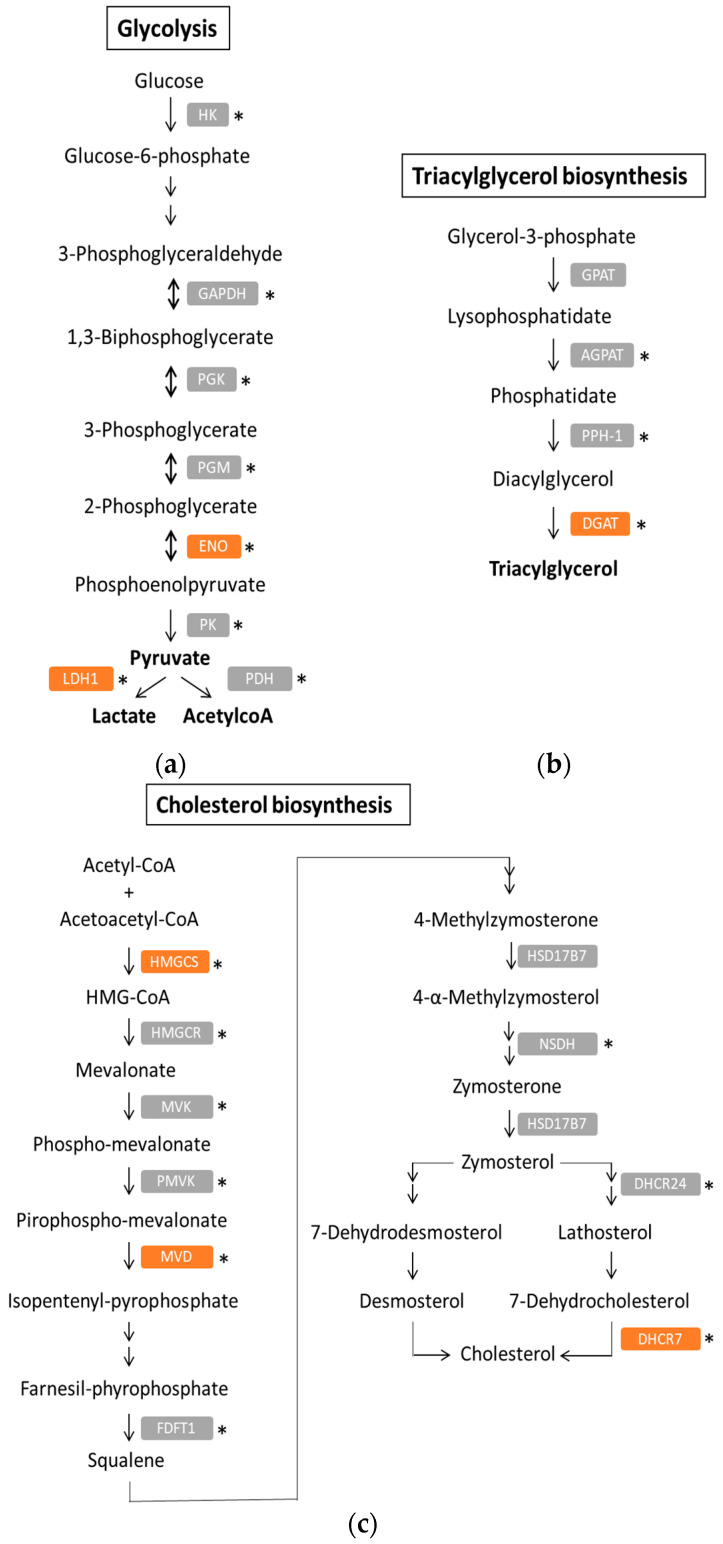
Genes from the glycolytic and lipogenic pathways up-regulated in PDAC samples. Expression of genes from glycolysis (**a**) triacylglycerol biosynthesis (**b**) and cholesterol biosynthesis (**c**) were analysed in cancerous (*n* = 179) and non-cancerous (*n* = 171) tissues (public gene expression information from PDAC vs. non-tumoural or normal pancreas tissues collected in the TCGA and GTEx databases) using GEPIA2 webtool (http://gepia2.cancer-pku.cn). The genes highlighted in the PDAC stratification studies from Daemen et al., Zhao et al. and Karasinska et al. are shown in orange. Significant differences in gene expression between cancerous and non-cancerous tissues was calculated by ANOVA test and is represented by an asterisk (* *p* ≤ 0.05).

**Figure 2 jcm-09-04128-f002:**
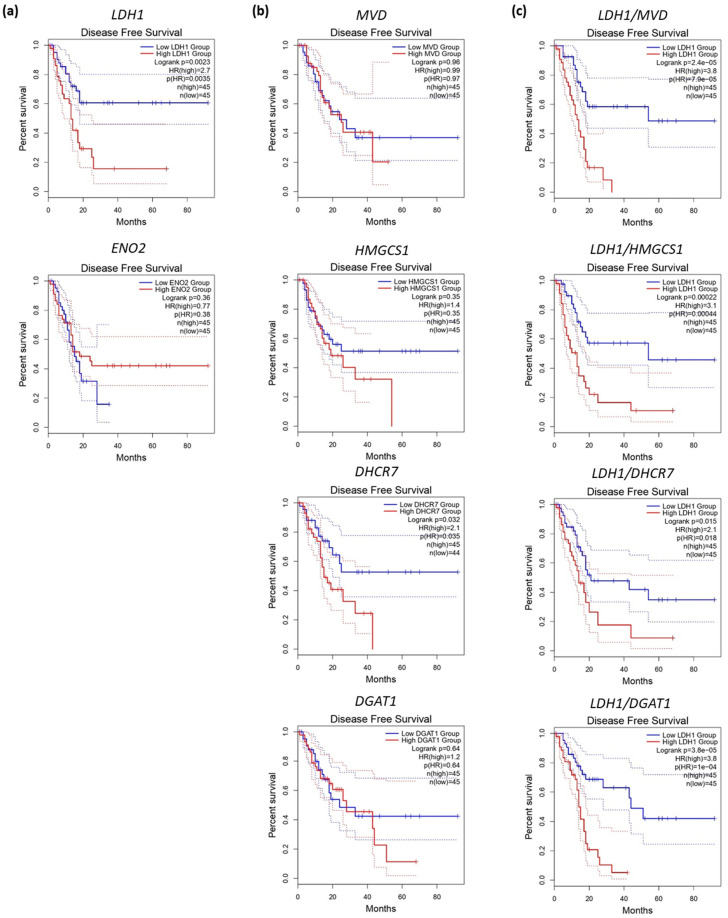
Disease-free survival analysis based on the expression of the genes selected for different metabolic classification systems. (**a**). *LDH1* and *ENO2*. (**b**) *MVD*, *HMGCS1*, *DHCRT* and *DGAT1*. (**c**) Ratio *LDH1* and *MVD*, *HMGCS1*, *DHCR7* or *DGAT1*. Comparison of upper and lower population quartiles is displayed. Log-rank test, Hazard Ratio (HR) and their corresponding *p*-values were calculated based on the COX model. Analyses were performed using the GEPIA2 web tool (http://gepia2.cancer-pku.cn) using public gene expression information collected in the TCGA database.

**Table 1 jcm-09-04128-t001:** Molecular Pancreatic Ductal Adenocarcinoma (PDAC) signatures.

Author	Type of Study	Type and Number of PDAC Samples	Dysregulated Pathways and Mutations	Outcome
Collison et al. [31]	Transcriptional	1. Clinical samples microarray datasets Microdissected (*n* = 27) GSE15471 (*n* = 36) GSE11838 (*n* = 107) GSE16515 (*n* = 52) E-MEXP-950 (*n* = 50) 2. Validation: Mouse cell lines (*n* = 15) Human cell lines (*n* = 19)	Classical: (↑) Adhesion-associated genes (*GATA6*). More K-RAS-dependent	Good
			Quasi-mesenchymal: (↑) Mesenchymal associated genes	Bad
			Exocrine: (↑) Digestive exocrine enzyme genes	
Moffit et al. [33]	Transcriptional	1. Microarray data Primary tumour (*n* = 145) Metastatic tumour (*n* = 61) Cell lines (*n* = 17) Pancreas normal samples (*n* = 46) Distant site adjacent samples (*n* = 88) 2. Validation Primary tumours (n = 15) PDXs (*n* = 37) Cell lines (*n* = 3) CAF lines (*n* = 6)	Classical: Classical Collison ((↑) adhesion-associated genes (*GATA6*)) and *SMAD4*	Good
			Basal: (↑) Genes also highly expressed in basal tumours in bladder and breast cancer	Bad
			Normal stroma: (↑) Pancreatic stellate cells, smooth muscle actin, vimentin and desmin markers	Good
			Activated stroma: (↑) Macrophages, tumour promotion and fibroblast activation-associated genes	Bad
Bailey et al. [10]	Mutational Transcriptional	Primary PDAC tumour samples and rare acinar cell carcinoma (*n* = 382) PDAC exomes (*n* = 74)	Squamous: Hypermethylation and (↓) pancreatic endodermal cell fate genes. *TP53*, *KDM6A* and *TP63*Δ*N*	Bad
			Pancreatic progenitor: (↑) Pancreatic early development genes (*PDX1*)	Good
			ADEX: (↑) K-RAS activation and pancreatic late development and differentiation genes	
			Immunogenic: (↑) Immune suppression and strong immune infiltration	
Zhao et al. [34]	Transcriptional (metanalysis)	1. Microarray datasets of PDAC primary tumour samples (*n* = 1268) TCGA (*n* = 172) GSE79670 (*n* = 51) TCGC PACA-AU (*n* = 71) MTAB-1791 (*n* = 195) ICGC array (*n* = 178) GSE71729 (*n* = 145) GSE62165 (*n* = 118) GSE62452 (*n* = 69) GSE57495 (*n* = 63) GSE60980 (*n* = 49) GSE77858 (*n* = 46) GSE55643 (*n* = 45) GSE15471 (*n* = 39)	L1: (↑) Metabolic genes	
			L2: (↑) Metabolic, cell proliferation and epithelium genes (*CDKN2A*)	Bad
			L3: (↑) Collagen and ECM associated genes	
			L4: (↑) Immune profile	Good
			L5: (↑) Neuroendocrine and insulin related pathways	Good
			L6: (↑) Metabolic and digestive enzyme genes	
Lomberk et al. [35]	Epigenetic	1. PDXs (*n* = 24) 2. Clinical samples microarray datasets GSE71729 (*n* = 145) ICGC (*n* = 178) TCGA (*n* = 172)	Classical: (↑) TFs involved in pancreatic development, metabolic regulators and Ras signalling	Good
			Basal: (↑) TF proliferative and transcription nodes	Bad
Maurer et al. [36]	Transcriptional Computational modelling	1. Primary PDAC tumour samples (*n* = 122) 2. Clinical samples microarray datasets GSE71729 (UNC) (*n* = 125) ICGC (*n* = 93) TCGA (*n*= 127)	Classical: Classical Moffit	Good
			Basal: Basal Moffit	Bad
			Immune-rich: (↑) immune and interleukin levels	Good
			ECM-rich: (↑) matrix extracellular pathways	Bad
Dijk et al. [37]	Transcriptional	1. Primary PDAC tumour samples (*n* = 90) 2. Pancreatic cancer PDXs cohort (*n* = 14) 3. PDAC Cell lines cohort (*n* = 51)	Epithelial: (↑) *MYC*, mitochondrial components and ribosome signature	Good
			Mesenchymal: (↑) *K-RAS*, pathways related to EMT, stromal signalling and TGF-β	Bad
			Compound pancreatic: Similar to the mesenchymal subtype and (↑) endocrine pathways	Good
Chan-Seng-Yue et al. [38]	Whole genome sequencing Transcriptional	Laser capture microdissected samples from late-stage PDAC 1. WGS (*n* = 314) 2. Bulk RNAseq (*n* = 248) 3. Single-cell RNAseq (*n* = 15)	Classical A/B: (↑) *SMAD4* and *GATA6* alterations	Good
			Basal-like A/B: (↑) EMT and TGF-β pathways, loss of *CDKN2A*, *TP53* mutations, *K-RAS* imbalance	Bad
			Hybrid	Mid
Nicolle et al. [39]	Transcriptional	PDXs (*n* = 76)	Graded types between classical and basal based on tumour differentiation	Grade dependant

For each classification, type of study, type and number of samples, dysregulated pathways and mutations and prognosis are described in each column. Up-regulated and down-regulated pathways are shown as (↑) and (↓), respectively. CAF, Cancer-Associated Fibroblast; EMT, Epithelial-to-Mesenchymal Transition; PDAC, Pancreatic Ductal Adenocarcinoma; PDX, Patient-Derived Xenograft; TF, Transcription Factor; WGS, Whole Genome Sequencing.

**Table 2 jcm-09-04128-t002:** Molecular subtypes corresponding to PDAC classification.

Authors	Common Subtypes			Others
Collisson et al. [31]	Classical	Quasi-mesenchymal	Exocrine-like	
Moffit et al. [33]	Classical	Basal-like		Normal and activated stroma
Bailey et al. [39]	Progenitor	Squamous	ADEX	Immunogenic
Zhao et al. [34]	L1	L2	L6	L3, L4 and L5
Lomberk et al. [35]	Classical	Basal		
Maurer et al. [36]	Classical	Basal		Immune-rich and ECM-rich
Dijk et al. [37]	Epithelial	Mesenchymal	Secretory	Compound pancreatic
Chan-Seng-Yue et al. [38]	Classical (A, B)	Basal-like (A, B)	Hybrid	
Nicolle et al. [39]	From Classical to Basal			

Correlated subtypes are grouped in columns. Colours define the subgroups with the best (green) or worst (red) prognosis. ECM, Extracellular Matrix.

**Table 3 jcm-09-04128-t003:** Metabolic PDAC signatures.

Author	Type of Study	Type and Number of PDAC Samples	Dysregulated Pathways, Metabolites and Mutations	Prognosis
Daemen et al. [92]	Metabolic Transcriptional	1. Metabolomic analysis Cell lines (*n* = 38) 2. Transcriptional analysis Cell lines (*n* = 38)	Slow-proliferating: (↓) amino acids and carbohydrates levels	
			Glycolytic: (↑) Metabolites and genes in glycolytic, pentose phosphate and serine pathways	Bad
			Lipogenic: (↑) Metabolites and genes in cholesterol and de novo lipid synthesis	Good
Zhao et al. [34]	Transcriptional	1. Microarray datasets of primary tumour samples (*n* = 1268) TCGA (*n* = 172) GSE79670 (*n* = 51) TCGC PACA-AU (*n* = 71) MTAB-1791 (*n* = 195) ICGCarray (*n* = 178) GSE71729 (*n* = 145) GSE62165 (*n* = 118) GSE62452 (*n* = 69) GSE57495 (*n* = 63) GSE60980 (*n* = 49) GSE77858 (*n* = 46) GSE55643 (*n* = 45) GSE15471 (*n* = 39)	L1: (↑) Glycolytic and lipogenic genes	
			L2: (↑) Glycolytic genes	Bad
			L3: (↑) Protein metabolism and digestive enzyme activity genes	
Lomberk et al. [35]	Epigenetic Transcriptional	1. PDXs (*n* = 24) 2. Clinical samples microarray datasets GSE71729 (*n* = 145) ICGC (*n* = 178) TCGA (*n*= 172)	Basal: (↑) MYC, glucose metabolism genes Classical: (↑) PPARs, lipid metabolism genes	Good
Maurer et al. [36]	Transcriptional Computational modelling	1. Primary PDAC tumour samples (*n* = 122) 2. Clinical samples microarray datasets GSE71729 (UNC) (*n* = 125) ICGC (*n* = 93) TCGA (*n* = 127)	Classical: (↑) lipogenic pathways (cholesterol, retinol and steroid hormone biosynthesis)	Good
Karasinska et al. [93]	Transcriptional Mutational	1. Transcriptional datasets TCGA (PAAD-US) (*n* = 61) ICGC (PACA-CA) (*n* = 148) COMPASS (*n* = 90) PanGen/POG (*n* = 26) 2. Mutational datasets TCGA (PAAD-US) (*n* = 60) ICGC (PACA-CA) (*n* = 86)	Quiescent: (↓) metabolic activity	
			Glycolytic: Glycolytic genes. *K-RAS* and *MYC* oncogenes amplification (↓) expression MPC1 and MPC2	Bad
			Cholesterogenic: (↑) Cholesterol biosynthesis genes	Good
			Mixed: (↑) Glycolytic and cholesterol biosynthesis genes	
Dijk et al. [37]	Transcriptional	1. Primary PDAC tumour samples (*n* = 90) 2. Pancreatic cancer PDXs cohort (*n* = 14) 3. PDAC Cell lines cohort (*n* = 51)	Epithelial: (↑) lipogenic pathways	Good

For each classification, type of study, type and number of samples, dysregulated pathways, metabolites and mutations and prognosis are described in each column. Up-regulated and down-regulated pathways are shown as (↑) and (↓), respectively.

**Table 4 jcm-09-04128-t004:** Schematic overview of the correspondence of reported molecular and metabolic subtypes in PDAC.

Authors	Subtypes	
Collisson et al. [31]	Classical	Quasi-mesenchymal
Moffit et al. [33]	Classical	Basal-like
Bailey et al. [39]	Progenitor	Squamous
Daemen et al. [91]	Lipogenic	Glycolytic
Zhao et al. [92]	L1 (Glycolytic/lipogenic)	L2 (Glycolytic)
Lomberk et al. [35]	Classical (PPAR-dep)	Basal (MYC/K-RAS dep)
Maurer et al. [36]	Classical (lipid metabolism)	Basal
Karasinska et al. [34]	Cholesterogenic	Glycolytic
Dijk et al. [37]	Epithelial (lipid metabolism)	Mesenchymal

Correlated subtypes are grouped in columns. Colours define the subgroups with the best (green) or worst (red) prognosis.

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
