# Peer review of "Molecular and Metabolic Subtypes Correspondence for Pancreatic Ductal Adenocarcinoma Classification"

_jcm, 2020, doi:10.3390/jcm9124128_

Round 1

Reviewer 1 Report

This review article is very detailed and well thought out. The authors have provided a very detailed classification of molecular and metabolic subtypes of PDAC and highlighted the possible prognostic values of this classification. 

   One of the major advantages of cancer type classification is the possibility of having targeted therapies. For example, EGFR targeted therapies in EGFR mutant cancers.

PDAC lacks any such targeted therapy for any specific driver mutation or metabolic subtype. The authors should discuss this problem with regard to PDAC subtype classification in the introduction and discussion section. 

Author Response

Thank you for this interesting suggestion. We have now included sentences to mention this problem in the introduction and at the end of section 5.

Reviewer 2 Report

In this review, the authors provide an interesting insight on the existing molecular and metabolic classifications of the PDAC tumours along with a detailed description mutation profiles. Subsequent to the discussion and analyses the authors propose a strategy to stratify patients using a metabolic signature taking into consideration genes representative of metabolic processes such as the glycolytic/lipogenic gene ratio from Daemen’s study that includes LDH1 instead of ENO2. They provide an interesting synthesis and highlight the importance of metabolic signatures in tumour aggressiveness and patient survival.

In my opinion this bibliographic work carried out by Espiau-Romera et al is of great interest. A significant effort has been made in recent years by the scientific community to classify pancreatic tumours with the aim of better stratifying patients for tailored therapies related to tumors with metabolic abnormalities. The literature in this area is now abundant and highly complex and further clarification on this topic is warrant. Unravelling this complexity and improving our understanding is paramount and in particular should be carried out by experts in the field such as P.Sancho’s group. Such efforts could significantly elucidate this perplexing but fascinating topic. Although many of the classifications were made from transcriptomic analysis, their relationship with metabolism is of high importance since metabolism provides better access to potential targets. I would like to congratulate the authors on the clarity of this manuscript and its relevance.

Minor comments:

1-I think it would be interesting to provide more detailed discussion related to the link between mutational status, molecular phenotypes and metabolic signatures.

2- In addition to the molecular phenotypes described in the review, it would be interesting to consider and discuss the recent findings by (M. Chan-Seng-Yue et al. Nature Genetics 2020, and N. Juiz et al. FASEB journal 2020) related to the existence of heterogeneous tumors containing more than one phenotype and associated with this, the possible evolution of PDAC driven by heterogeneity, the mutational status and genomics instability.

3- how the gene expression level of lactate dehydrogenase 1 (LDH1) could impact patient survival?

Author Response

1-I think it would be interesting to provide more detailed discussion related to the link between mutational status, molecular phenotypes and metabolic signatures.

Thank you for this valuable suggestion. We have now included new sentences regarding this link at the end of section 5.

2- In addition to the molecular phenotypes described in the review, it would be interesting to consider and discuss the recent findings by (M. Chan-Seng-Yue et al. Nature Genetics 2020, and N. Juiz et al. FASEB journal 2020) related to the existence of heterogeneous tumors containing more than one phenotype and associated with this, the possible evolution of PDAC driven by heterogeneity, the mutational status and genomics instability.

We would like to thank the reviewer for this suggestion and apologize for the omission of such crucial findings for our study. We have now included the work by Chan-Seng-Yue et al in our classification and briefly discussed about the heterogeneity of PDAC tumors (tables 1 and 2, and text in section 3) resolved at single cell level (final paragraph of section 3).

3- how the gene expression level of lactate dehydrogenase 1 (LDH1) could impact patient survival?

The impact of LDH1 on patient survival probably reflects the increased malignant behaviour of glycolytic tumors, since LDH1 is the most representative gene of this pathway.